# Bacterial Protein Tyrosine Phosphatases as Possible Targets for Antimicrobial Therapies in Response to Antibiotic Resistance

**DOI:** 10.3390/antiox11122397

**Published:** 2022-12-02

**Authors:** Alicja Kuban-Jankowska, Tomasz Kostrzewa, Magdalena Gorska-Ponikowska

**Affiliations:** 1Department of Medical Chemistry, Medical University of Gdansk, 80-210 Gdansk, Poland; 2Euro-Mediterranean Institute of Science and Technology, 90139 Palermo, Italy; 3Department of Biophysics, Institute of Biomaterials and Biomolecular Systems, University of Stuttgart, 70174 Stuttgart, Germany

**Keywords:** protein tyrosine phosphatase, reversible oxidation, bacterial phosphatase, virulence factor, antimicrobial therapy

## Abstract

The review is focused on the bacterial protein tyrosine phosphatases (PTPs) utilized by bacteria as virulence factors necessary for pathogenicity. The inhibition of bacterial PTPs could contribute to the arrest of the bacterial infection process. This mechanism could be utilized in the design of antimicrobial therapy as adjuvants to antibiotics. The review summaries knowledge on pathogenic bacterial protein tyrosine phosphatases (PTPs) involved in infection process, such as: PTPA and PTPB from *Staphylococcus aureus* and *Mycobacterium tuberculosis*; SptP from *Salmonella typhimurium*; YopH from *Yersinia* sp. and TbpA from *Pseudomonas aeruginosa*. The review focuses also on the potential inhibitory compounds of bacterial virulence factors and inhibitory mechanisms such as the reversible oxidation of tyrosine phosphatases.

## 1. Introduction

Recently, bacterial pathogens have posed a serious threat to human health worldwide. Intensified research on bacterial pathogenesis over the last decades greatly expanded our knowledge of the mechanisms of disease processes at the molecular level. However, modern medicine is still not always able to cope with all cases. Increasing antibiotic-resistant, emerging strains and re-emerging infectious agents have become increasingly common and present enormous challenges for treating infections caused by highly resistant bacterial strains. In the process of evolution, bacteria develop mechanisms that block or bypass the pathways of the action of commonly used antibacterial drugs. For example, the MRSA strain of *Staphylococcus aureus* is resistant to all β-lactam antibiotics by producing the β-lactamase enzyme. Strains of bacteria resistant to all known antibiotics have already appeared, e.g., *K. pneumoniae* New Delhi metallo-β-lactamase (NDM) [1].

This raises the need for a new approach to the topic of treatment and prevention of bacterial infections using non-standard and more effective antimicrobial compounds. As an answer to the growing resistance of pathogenic bacteria to antibiotic therapy, scientists need to design novel promising antimicrobial compounds that can be utilized as an addition to standard therapy, thereby increasing its effectiveness. The novelty is to target directly secreted virulence factors controlling infection processes.

Due to antibiotic resistance, finding new ways to fight bacteria that are difficult to treat is now a real challenge. Numerous research studies are dedicated to the search for inhibitors of virulence factors, including bacterial tyrosine phosphatases.

The main goal for scientific groups is the selection and identification of suitable inhibitory compounds against pathogenic bacterial protein tyrosine phosphatases (PTPs) involved in the infection process, such as PTPA and PTPB from *Staphylococcus aureus* and *Mycobacterium tuberculosis*; SptP from *Salmonella typhimurium*; YopH from *Yersinia* spp. and TbpA from *Pseudomonas aeruginosa*.

This review lists selected antimicrobial compounds—inhibitors of the bacterial protein tyrosine phosphatases used as a virulence factors necessary for induction of infection. The inhibition of bacterial PTPs could contribute to the arrest of the bacterial infection process. It could be utilized in the design of modern antimicrobial therapy or as a supportive therapy for standard treatment.

## 2. Bacterial Pathogenicity and Tyrosine Phosphorylation

One of the many mechanisms of bacterial virulence is the activity of bacterial kinases and phosphatases. As in eukaryotic cells, in bacteria a lot diverse families of enzymes with this type of activity can be found. These enzymes phosphorylate and dephosphorylate various amino acid residues in proteins, most often serine (Ser), threonine (Thr), tyrosine (Tyr), histidine (His) and arginine (Arg). Phosphorylation of the specific amino acids in proteins is an essential component of many signaling pathways. In addition to protein kinases and phosphatases, an important role is played also by phosphoproteins that capture other regulatory proteins. Reversible phosphorylation and dephosphorylation can control the activity of target proteins, either directly through conformational changes in proteins, or indirectly by regulating protein–protein interactions [2].

Reversible phosphorylation of structural and regulatory proteins is an important control mechanism in eukaryotic cells. Protein tyrosine phosphatases regulate signal transduction pathways that control numerous cell functions, including proliferation, differentiation and growth [3,4]. An attempt to study the mechanisms of the regulation of protein tyrosine phosphatases, which are modulators cell signaling, and thus the numerous functions of cells, seems to continue an important research goal. The malfunction of these phosphatases is associated with numerous pathologies, including neoplastic processes [5].

Protein tyrosine phosphorylation has emerged as a key tool in the control of numerous cellular functions in bacteria. Numerous bacteria utilize secreted protein tyrosine phosphatase activity to take an essential part in host–pathogen interactions involved with the manipulation of host signaling pathways and, subsequently, with the induction of infection process. Bacterial tyrosine phosphorylation/dephosphorylation is also involved in biofilm formation and community development [6].

Bacterial protein tyrosine phosphatases and tyrosine phosphorylation represent critical mechanisms of the control of crucial processes of pathogenic bacteria. Some of those PTPs are involved in the regulation of biosynthesis of capsular and extracellular polysaccharides, critical for bacterial virulence [7]. Bacterial PTPs became a promising target for the design and development of novel antimicrobial compounds, including small molecule inhibitors from natural sources [8].

## 3. Bacterial Tyrosine Phosphatases—Bacterial Virulence Factors

Recently, rapid progress in bacterial genome sequencing has led to the discovery and characterization of numerous novel virulence factors. One of the many mechanisms in bacterial infection process is the activity of bacterial kinases and phosphatases. These enzymes phosphorylate and dephosphorylate amino acid residues in proteins, most commonly serine, tyrosine or threonine. Reversible phosphorylation and dephosphorylation control the activity of target proteins by induction of conformational changes or by regulation of protein–protein interactions.

The virulence of microorganisms is the ability of the pathogen to penetrate, replicate, multiply and consequently damage the tissues of the infected organism [9]. The virulence determinants of a pathogen are any of its genetic, biochemical or structural features that enable it to cause disease in the host. In recent years, rapid progress in bacterial genomic sequencing has led to the discovery and characterization of many new virulence factors. Some of the identified virulence factors help the bacteria to adapt physiologically and metabolically in a hostile environment, while others are secreted and carry out a series of biological and immunological modulations [10].

Secretory molecules responsible for the virulence of the strain are located in the cell membrane or in the cytosol. Secretory factors are important ingredients that help the bacteria get through the innate and adaptive response of the host immune system. Membrane-bound virulence factors promote adhesion of bacteria and penetration into the infected cell. Cytosolic factors facilitate rapid adaptation of the bacteria in the host organism—metabolic, physiological and morphological changes [11]. The virulence of microorganisms is determined by secreted proteins, such as protein toxins and enzymes, as well as structures associated with the cell surface, including polysaccharides, lipopolysaccharides and outer membrane proteins, which directly contribute to disease processes. Many coding genes virulence traits are also indirectly involved in pathogenesis [12].

A characteristic phenomenon used by some virulent bacterial strains is the secretion of virulence factors into the interior of infected cells, which by changing the signaling pathways enable initiation of the infection process. These virulence factors include bacterial protein tyrosine phosphatases whose activity is necessary for the total bacterial virulence [13].

### 3.1. Virulence Factors from Staphylococcus aureus and Mycobacterium tuberculosis

*Staphylococcus aureus* is a Gram-positive bacterium that can commonly occur on the skin and mucosa of the nasal cavity in humans, usually without causing symptoms [14]. However, *S. aureus* can cause a wide range of ailments, from skin and soft tissue infections, through pneumonia, osteomyelitis, to sepsis [15]. There is also a growing group of *Staphylococcus aureus* strains that have produced resistance to available, strong antibiotics—MRSA resistant to β-lactams, macrolides and fluoroquinolones [16] and vancomycin-resistant VRSA strain [17].

*Mycobacterium tuberculosis*, the etiological agent of tuberculosis, is acidophilic bacteria, resistant to many environmental factors, such as drying, high and low temperature, high and low pH. Tuberculosis can be divided into pulmonary form and extrapulmonary. The most common, pulmonary, manifests itself mainly with a long-lasting cough. The extrapulmonary form mainly affects people with reduced immunity and affects the pleura, lymph nodes, bones or the urinary system [18]. Treatment is based on a total of 6 months of combination therapy with four drugs from different classes (isoniazid, rifampicin, pyrazinamide and ethambutol). Furthermore, in the case of *Mycobacterium tuberculosis*, there are cases of difficult-to-treat rifampicin-resistant strains, requiring prolonged and intensified therapy [19].

*Staphylococcus aureus* and *Mycobacterium tuberculosis* bacteria produce two low molecular weight tyrosine phosphatases: PtpA and PtpB. Crystallographic studies have shown a similarity of the PtpA and PtpB sequences with the low molecular weight protein tyrosine phosphatases (LMW-PTPs) family; however, their topology differs. First of all, the P-loop motif in the active site is located in the N-terminal domain of the protein, in contrast to classical and dual specific tyrosine phosphatases where this loop is located in the middle of the sequence. This difference suggests that LMW-PTPs evolved separately from classical and dual-specific PTPs [20,21].

PtpA phosphatase from *Staphylococcus aureus* is secreted during macrophage infection and has a critical role as a bacterial effector protein that prevents host defense [22]. PtpA phosphatase is secreted during growth of *S. aureus*, and deletion of PtpA influences on survival and infectivity [23].

*Mycobacterium tuberculosis* bacteria secrete PTPA and PTPB phosphatases into the cytosol of the macrophage cell, where it leads to the destruction of the key components of the endocytic pathway, consequently leading to the arrest of phagosome maturation [24].

### 3.2. SptP from Salmonella typhimurium

*Salmonella typhimurium* belongs to Gram-negative rods of the *Salmonella* family. It is one of the etiological factors of salmonellosis. The disease manifests itself in varying degrees of gastroenteritis. It usually resolves spontaneously, does not require causal treatment, only symptomatic. Antibiotic therapy based on fluoroquinolones or macrolides is used only in severe course or parenteral infection [25].

*Salmonella typhimurium* interacts with host cells to stimulate signaling pathways, leading to a variety of cellular responses, including cytoskeleton rearrangement, cytokine production and, in some cell types, programmed cell death or apoptosis. This interaction is largely dependent on the function of proteins located at position 63 of the *Salmonella* chromosome. This type of protein secretion system has been identified in many Gram-negative bacteria pathogenic to animals and plants that share the ability to engage host cells in complex interactions. It is generally assumed that the main function of this mechanism is the translocation of bacterial proteins into the host cell, which can then stimulate or disrupt host cell signaling pathways [26].

*Salmonella typhimurium* bacteria utilize the type III secretion system and translocate SptP phosphatase to epithelial cells, where it inhibits mitogen-activated kinases that play a role in regulating the response to external signals reaching the cell by altering gene expression or apoptosis of cells [27].

SptP phosphatase has been identified as an effector protein in *Salmonella typhimurium*. SptP has a modular structural organization that may reflect the presence of different effector domains. SptP is deployed in modular domains. The N-terminus shares sequence similarity with two other bacterial toxins secreted by functionally homologous type III secretion systems: *Yersinia* YopE and *Pseudomonas* ExoS. The C-terminal domain, on the other hand, is similar to the YopH phosphatase. The probable contribution of SptP phosphatase in *Salmonella* infection process include the change in cell physiology, reorganization of the cytoskeleton and entry and survival in host tissues [28].

### 3.3. Pathogenicity of Pseudomonas aeruginosa

*Pseudomonas aeruginosa* is a bacterium that shows great metabolic diversity. It is found in many biotic and abiotic habitats, including water, soil and various organisms. In humans, it is responsible for opportunistic infections. It is a cause of often dangerous infections in people with lowered immunity for various reasons, patients with cystic fibrosis and those who are mechanically ventilated. Most often, it causes infections of the respiratory system, the middle ear (so-called swimmer’s ear) and the urinary system (especially during long-term catheterization). *P. aeruginosa* is resistant to most antibiotics, and hospital strains are often sensitive only to very strong antibiotics such as aztreonam and colistin [29].

The versatility of this bacterium is related to the large number of regulatory proteins in its genome. Due to the membrane permeability barrier, *Pseudomonas aeruginosa* has acquired a high level of drug resistance, which makes the treatment of patients infected with this pathogen extremely difficult. Critical features that contribute to the pathogenicity of *Pseudomonas aeruginosa* include the production of multiple virulence factors, biofilm formation and antibiotic resistance [30].

*P. aeruginosa* is one of the most dangerous microorganisms causing nosocomial infections, the treatment of which is very difficult due to the high resistance of this bacterium to antibiotics. Critical characteristics that contribute towards the pathogenicity of *P. aeruginosa* include the production of virulence factors and biofilms [31,32]. The formation of biofilm is crucial for persistence in host cells, development of chronic infections and for cell growth and communication [33]. TpbA from *P. aeruginosa* is secreted to the periplasm and connected with the extracellular quorum, sensing signals to extracellular polysaccharide production and biofilm formation [34]. Inactivation of TpbA causes wrinkled colony morphology, which is related to cell persistence in clinical infections. TpbA mutation results in undeveloped biofilm structures, as the mutation forms flat and thick biofilms [35]. In addition, TpbA phosphatase binds transferrin on its surface and participates in the acquisition of iron necessary for the functioning of bacteria [36].

### 3.4. Yops phosphatases as Virulence Factors from Yersinia

*Yersinia* genus contains three species of bacteria pathogenic to humans: *Yersinia pestis* is the etiological agent of plague; *Yersinia pseudotuberculosis* infection is most often manifested by mesenteric lymphadenitis and *Yersinia enterocolitica*, which is responsible for a number of gastrointestinal disorders and lymphadenitis. The bacteria of the genius *Yersinia* are the one of the most virulent pathogens threatening humans, being the cause of many epidemics in the past and still posing a threat through the animal reservoirs or by danger of using them as a biological weapon due to their high virulence. There are still many human cases caused by *Yersinia pestis*, with the greatest frequency of human plague infections having occurred in Africa [37]. Moreover, the presence of *Y. pestis* in wild reservoir animals (i.e., from national parks) is detected also in highly developed countries [38]. *Yersinia* genus represents the species of bacteria pathogenic to humans: plague-causing *Y. pestis*, *Y. pseudotuberculosis* inducing tuberculosis-like symptoms and septicemia or *Y. enterocolitica* responsible for gastrointestinal disorders [39]. *Y. pestis* is transmitted through blood by fleas from its natural reservoirs, mainly rodents, squirrels, chipmunks or rabbits, and leads to the bubonic form of plague [40]. The inhalation of the infectious respiratory droplets of these bacteria results in the most severe primary pneumonic plague, with mortality rates approaching 100 percent in the absence of treatment [41]. Both forms can lead to infection of the blood, causing bacteremia and septicemic plague.

Bacteria *Yersinia* spp. use the type III secretion system to translocate virulence effectors deep into the host cell. During infection, *Yersinia* translocates Yop virulence effectors (Yops) into the host cell, leading to the suppression of the innate immune response [42]. Numerous virulence effectors of *Yersinia* bacteria are phosphatases that are crucial during the infection process (Figure 1). One of the effectors of *Yersinia* sp. outer membrane proteins is the highly active protein tyrosine phosphatase YopH, which is an essential virulence factor for bacteria. YopH deregulates host cellular functions and blocks phagocytosis. In addition, YopH, by dephosphorylation of focal adhesion kinase (FAK), prevents host cell adhesion and inhibits the production of reactive oxygen species by macrophages. The YopH catalytic site contains an amino acid sequence that is similar to the eukaryotic protein tyrosine phosphatase. There is a cysteine residue in the active site that is essential for catalysis and enzymatic activity [39].

The secretion model for translocation leads to secretion of virulence factors, among others YopH protein tyrosine phosphatase, which is involved in the inhibition of phagocytosis used by macrophages to get rid of pathogens [42,43].

## 4. Inhibition of Bacterial Protein Tyrosine Phosphatases

Due to the problem of antibiotic resistance, new ways to fight bacteria that are difficult to treat are currently being sought. Many studies are devoted to the search for inhibitors of virulence factors such as tyrosine phosphatases, which, by regulating them, could become an interesting option supporting current treatments. The phosphorylation of proteins containing tyrosine residues is a key post-translational modification controlling numerous cellular functions of bacteria. So far, a number of tyrosine phosphatases have been found to be responsible for the virulence of various bacterial strains. Many species of bacteria use the activity of protein tyrosine phosphatase in the host–pathogen interaction, influencing signaling pathways and subsequent induction of the infection process. A lot of work has been devoted to the search for tyrosine phosphatase inhibitors in the context of possible support of the current antibacterial treatment.

### Reversible Oxidation as a Possible Inhibitory Mechanism of Tyrosine Phosphatases

PTPs serve as key mediators of cell signaling and the reactive oxygen species (ROS) level, as reversible oxidation is the main mechanism for control of PTPs’ activity. The catalytic cysteine residue in the active center of protein tyrosine phosphatases is in the form of a thiolate anion, and because of its microenvironment has a low pKa (~5.4), which induces sensitivity to oxidation. The oxidation of catalytic cysteine inhibits the ability of the enzyme to dephosphorylate the substrate, involving the transfer of a phosphate group from the substrate to the catalytic cysteine. Depending on the degree of oxidation, the cysteine residue in the active center may turn into a form of sulfenic (SOH), sulfinic (SO_2_H) or sulfonic acid (SO_3_H) Figure 2. Oxidation of cysteine residues to form a sulfenic acid is reversible, because it can be reduced in a cell by glutathione or thioredoxin. Accordingly, the high oxidation potential of the environment can result in the conversion of cysteine to an irreversible form of sulfinic or sulfonic acid [44]. However, in some cases, sulfiredoxin (*S. cerevisiae* yeast protein) can reduce sulfinic acid residue of peroxiredoxin Tsa1 in the presence of magnesium ions and ATP [45]. Peroxiredoxin, like tyrosine phosphatases, contains the redox-active cysteine residue in the active site [46].

The reversible oxidation of catalytic cysteine residue is a characteristic mechanism of the regulation of protein tyrosine phosphatases. The oxidized form can be restored to an active reduced form by converting a sulfenic acid to a transitional structure of sulfenylamide. The close localization of the histidine and cysteine residues in this structure causes the polarity of the amide bond, enabling nucleophilic attack of the nitrogen atom of the serine residue at the sulfur atom of a cysteine residue in sulfenic acid. This leads to condensation and formation of a covalent bond between the sulfur and nitrogen atoms. Sulfenylamide may then be reduced to the active form of the thiolate anion [47]. The formation of sulfenylamide induces a conformational change in the catalytic center of the enzyme, which protects the catalytic cysteine from irreversible oxidation to sulfinic acid and sulfonic acids, and facilitates the reactivation of the enzyme by the action of biological reducing agents, i.e., thioredoxin or glutathione [48].

## 5. Potential Bacterial PTP Inhibitors

The overwhelming majority of the known tyrosine phosphatase inhibitors are inorganic compounds, i.e., sodium orthovanadate, nitric oxide or phenyl arsine oxide. The main disadvantage of these compounds is that they are not specific and may cause an effect on a wide spectrum of human proteins. Numerous tyrosine phosphatases inhibitors described so far may not possess drug-like characteristics and have low cell permeability and selectivity [49]. There are many studies focused on designing novel, more potent and selective PTP inhibitors. The main problem for all of these studies is the high degree of similarity in the catalytic site of all protein tyrosine phosphatases [50]. The identification of a novel binding site of PTPs located about 20 angstroms from a catalytic center, which is less conserved among phosphatases, constitutes a new paradigm for inhibitor design. Both the catalytic center and the secondary binding site may bind to the potential inhibitory substances, which may cause loss of enzymatic activity or allosteric inhibition [51].

Thanks to the latest research studies, it has already been possible to design a few inhibitors with high activity against tyrosine phosphatases (Table 1). The search for new compounds that would be effective against bacterial virulence factors is now a challenge for modern medicine and could be an alternative or support to antibiotics. Regarding the problem of the mutation of bacterial strains through which the mechanism of resistance is developed in the drugs, a key aspect of currently conducted research is the search for new methods of therapy.

### 5.1. Oxidants as PTP Inhibitors

It is believed that the reversible oxidation of catalytic cysteine can be a universal mechanism for regulating the activity of protein tyrosine phosphatases; thus, all compounds that possess oxidizing properties are believed to act as PTP inhibitors.

Several studies have demonstrated that the enzymatic activity of protein tyrosine phosphatases depends on the level of oxidative stress. It was found that the stress factors—i.e., hydrogen peroxide, UV light, as well as the thermal shock—result in the inactivation of the receptor protein phosphatase RPTPα [61]. The inactivation of a number of protein tyrosine phosphatases, including by hydrogen peroxide, were also demonstrated [62]. Hydrogen peroxide is a factor that may cause reversible oxidation of catalytic cysteine residues to a sulfenic acid and a loss of the enzyme activity, which in laboratory conditions can be restored by thiol reducers, i.e., dithiothreitol [63,64]. However, prolonged exposure of the catalytic cysteine residues to hydrogen peroxide can lead to the irreversible oxidation [65,66].

The more powerful oxidants that can be generated in the presence of hydrogen peroxide and carboxylic acids are peracids. Peracids are highly reactive and can induce inactivation of thiol groups containing enzymes [67]. The ability to inhibit protein tyrosine phosphatases was studied by our group in previous years. We discovered that peracids are also able to inhibit bacterial protein tyrosine phosphatase, YopH, with an IC50 value of 41 nM observed for peroxyctanoic acid (C8-peracid) against YopH from *Yersinia pestis* [59] and YopH from *Yersinia enterocolitica* [60].

Not only oxidants can induce inhibition of PTPs via oxidation mechanism, but ROS generating compounds such as aurintricarboxylic acid (ATA) can as well. ATA is believed to be a nontoxic compound, and it is still one of the most active inhibitors of the YopH virulence factor from *Yersinia* bacteria, with IC50 equal to 10 nM (presented in Table 1) [56].

### 5.2. Inhibitors from Natural Sources

There are many natural compounds that inhibit different types of enzymes, including protein tyrosine phosphatases. Some natural PTP inhibitors can be extracted from plants, algae or microorganisms. For example, dephostatin has been reported to be a competitive PTP inhibitor in micromolar doses, isolated from *Streptomyces* spp. [68]. Another inhibitor, 4-isoavenaciolide, was extracted from a fungal strain. There are also plenty of compounds from fruits found to possess inhibitory properties against PTPs, i.e., nornuciferine from *Annona muricata* or karanjin from *Pongamia pinnata*. The therapeutic utilization of natural compounds is partially limited by their low stability and selectivity, but they can be used in the design of synthetic analogues [50].

Bacterial PTPs became a promising target for the design and development of novel antimicrobial compounds including small molecule inhibitors from natural sources [6]. Mascarello and his team presented six natural compounds proven to inhibit PtpB at low micromolar concentrations (<30 µM), with Kuwanol E being the most potent with Ki = 1.6 ± 0.1 µM [54]. The promising inhibitory compounds are those with Ki below 1 µM.

### 5.3. Inhibitors That Mimics PTPs’ Natural Substrates

Many studies involving NMR-based screening or molecular modeling focused also on a search for the compounds that inactivate protein tyrosine phosphatases by mimicking the natural substrate of PTPs: phosphotyrosine. The phosphotyrosyl in mimetic structures is replaced by sulfotyrosyl, thiophosphotyrosyl or phosphonomethylphenylalanine [69]. Most inhibitory compounds that are phosphotyrosine mimetics, despite the high inhibitory activity, are characterized by relatively low permeability through the cell membrane. The compounds that mimics natural substrates could be also promising bacterial phosphatases inhibitors [20].

## 6. Conclusions and Further Remarks

Pharmacists and scientists have focused their attention on the regulatory mechanisms of the activity of protein tyrosine phosphatases because of the relationship of the development of human diseases with impaired activity of phosphatases. Protein tyrosine phosphatases become potential pharmacological targets for the design and production of drugs, including the discovery of natural substances that may act as phosphatase inhibitors [68,70]. The modulation of enzyme activity by using PTP inhibitors might play an important role in pharmacology [50].

Protein phosphorylation of tyrosine residues appears to be a key tool in the regulation of cellular and physiological processes in both eukaryotic and prokaryotic cells. Impressive progress has been made in recent years in the identification of protein kinases and protein phosphatases, which are responsible for bacterial virulence. This review is focused on bacterial tyrosine phosphates; however, there are numerous bacterial tyrosine kinases that play essential roles in bacterial virulence. An example of such a kinase is CapB2 from Staphylococcus aureus, which is responsible together with PtpA and PtpB tyrosine phosphatases for the production of polysaccharide capsule [71].

Due to the knowledge of inhibitory mechanisms, it is possible to design strong potential inhibitors drugs that, by interacting with the active sites of tyrosine phosphatases, can provide both primary and adjuvant treatment.

Based on reviewed studies, it can be concluded that potential inhibitors of bacterial tyrosine phosphatases are mainly compounds with low molar mass, of an electrophilic nature, with carboxyl groups that will form intermolecular hydrogen interactions and rich in aromatic rings that would guarantee the interaction of pi stacking with side chains of phosphatase-building amino acids.

Many important questions still remain regarding, in particular, the nature of the effectors that switch on and switch off the relevant networks and the cascades of reactions taking place throughout the process of phosphorylation/dephosphorylation. There is also a gap between the growing number of those identified protein-tyrosine kinases and phosphatases and the number of designed or identified potential inhibitors. Therefore, the search for new compounds that inhibit tyrosine phosphatases seems to result in the development of new therapeutic strategies, which help in the fight against strains resistant to antibiotics and difficult bacterial infections.

## Figures and Tables

**Figure 1 antioxidants-11-02397-f001:**
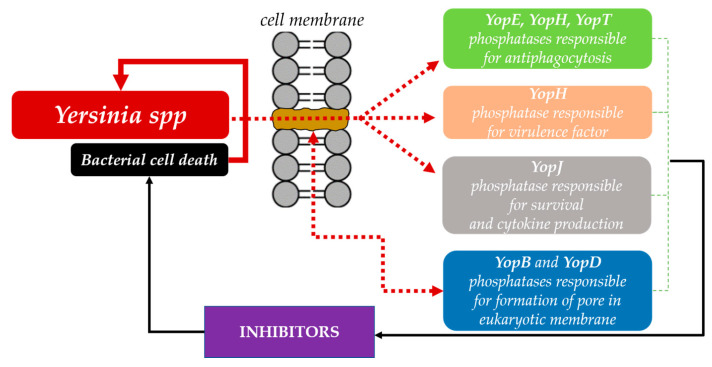
Phosphatases as virulence factors of *Yersinia* spp.

**Figure 2 antioxidants-11-02397-f002:**
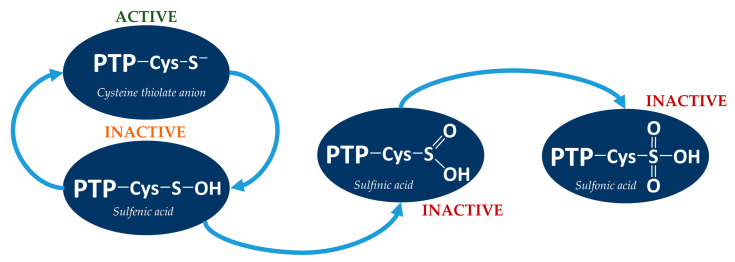
Inactivation of tyrosine phosphatase by oxidation of the catalytic cysteine residue to sufenic, sulfinic or sulfonic acid.

**Table 1 antioxidants-11-02397-t001:** Structure of bacterial protein tyrosine phosphatase inhibitors.

Inhibitor Structure	Phosphatase	IC_50_	References
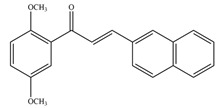	PtpA	8.4 ± 0.9 µM	[52]
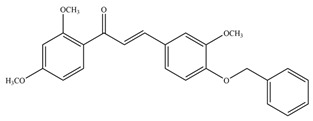	PtpA	15.0 ± 4.0 µM	[53]
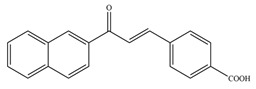	PtpB	12.0 ± 2.0 µM	[53]
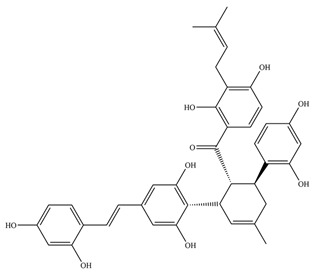	PtpB	1.9 ± 0.5 µM	[54]
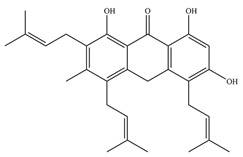	PtpB	5.4 ± 0.6 µM	[54]
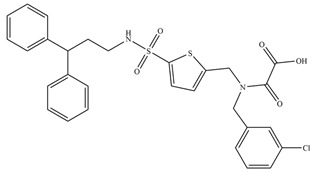	PtpB	440 ± 50 nM	[55]
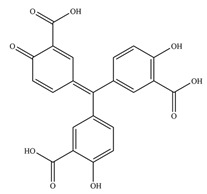	YopH	10 ± 2.0 nM	[56,57]
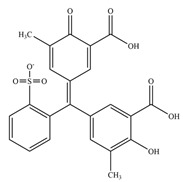	YopH	59.5 ± 6.2 µM	[56]
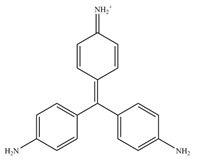	YopH	79.1 ± 9.7 µM	[57]
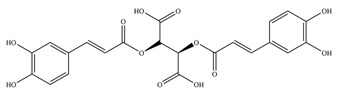	YopH	250 µM	[49,58]
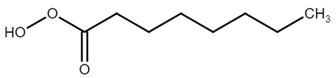	YopH *Y. pestis*	41 nM	[59]
	YopH *Y. enterocolitica*	39 nM	[60]

## Data Availability

Not applicable.

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
