# Peer review of "Bacterial Protein Tyrosine Phosphatases as Possible Targets for Antimicrobial Therapies in Response to Antibiotic Resistance"

_antioxidants, 2022, doi:10.3390/antiox11122397_

Round 1
Reviewer 1 Report
Dear authors
I completed the revision of article entitled “Bacterial protein tyrosine phosphatases - new possible targets for antimicrobial therapies”
Herein, you can fin my comments.
This review deals with a very important and current aspect, which is the development of new antimicrobial drugs that target bacterial PTPs. These enzymes are recognized as important virulence factors that regulate the host infection process of numerous human pathogenic bacteria.
The authors provided an overview of the functions of PTPs and then focused on the role of some bacterial PTPs. In particular, they focused their attention on PTPs produced by Staphylococcus aureus, Mycobacterium tuberculosis, Salmonella typhimurium, Yersinia spp. and by Pseudomonas aeruginosa.
Furthermore, the authors described the role of reversible oxidation as a mechanism for regulating PTP activity and highlighted the difficulties associated with the production of new and specific bacterial PTP inhibitors. As an example, the authors showed a number of natural molecules capable of acting as inhibitors of bacterial PTPs.
Finally, the authors concluded their manuscript by underlining the role that this type of molecule could have in the development of new drugs capable of fighting bacterial infections.
Overall, I consider this topic very interesting.
However, I think this manuscript is too generic, and badly organized. Furthermore, the content of the manuscript does not meet the expectations generated in the reader by the title which suggests the identification of new potential targets for the treatment of bacterial infections. In reality, as we can deduce from the literature, the targets described are not innovative at all. Rather, the proposed inhibitors could be. However, even in this case, it was not clear why the proposed molecules should represent an innovation. Furthermore, the part concerning the oxidants appears, as presented, out of place. Rather than describing the role of hydrogen peroxide as an oxidant / inhibitor of PTPs, a well known evidence, the authors should discuss the possibility of generating new inhibitors capable of acting through a redox mechanism. Are there any known inhibitors that act in this way? Is their use desirable against bacterial PTPases?
For these reasons, I think that this manuscript is not ready for publication in the journal Antioxidant.
Best regards
Author Response
Thank you so much for all the valuable comments and suggestions.
We reorganized the manuscript.
We added a lot more material to avoid the paper to be too generic.
We changed the title.
We included the section about oxidation of PTPs as it is very important mechanism of regulation its activity, we extended the part about the inhibitors that act as oxidants and add ROS generating compound that inhibit activity of bacterial tyrosine phosphatases. The good example of such a compound is aurintricarboxylic acid, as it is strong and nontoxic inhibitor (Table 1 , ref: 56).
Reviewer 2 Report
This nice and very focused review addresses the role and possible inhibition of bacterial protein tyrosine phosphatases (PTPs), which are important for bacterial pathogenicity. This is a very important, often neglected topic in biomedical and antimicrobial research. The authors have published interesting papers in this area and are therefore highly qualified for the present review. This referee only has some comments / suggestions, which the authors may want to address.
1) The review concentrates on PTPA/PTPB from Staphylococcus aureus and Mycobacterium tuberculosis; SptP from Salmonella typhimurium; YopH from Yersinia sp. and TbpA from Pseudomonas. aeruginosa. Especially for non-bacteriologists it would be of high interest if the authors expand the section describing the molecular properties of these phosphatases. Is the Y-protein phosphatase activity essential for the biological activity? Could they have function as adaptor proteins? Are the most important substrates Y-phosphorylated host proteins, which ones? This referee suggests to delete Fig.1 (too general) and, instead, expand Figure 2/ text to the other phosphatases (see also point 3).
2) The authors discuss the interesting finding that a reversible oxidation of a catalytic cysteine can be an important mechanism of regulation including inhibition. Interestingly, there are also important cysteines in the catalytic center of protein tyrosine kinases such as Tec/Btk tyrosine protein kinases (here a cysteine 481), which are targeted by protein kinase inhibitors. Have these Cys-containing sites compared at the molecular level ?
3) It is mentioned that there was impressive progress in studying bacterial tyrosine protein kinases. Are these bacterial TKs the direct counterparts of the tyrosine phosphatases reviewed here or others, such as eukaryotic enzymes? Perhaps, a very shot overview at the beginning of this review would be helpful.
Author Response
Thank you so much for all the valuable comments and suggestions.
We extend the section about bacterial phosphatases as virulence factors. We deleted Fig.1 as suggested.
There is not such a comparison between Cys-containing kinases and phosphatases, as Cys must be surrounded by other essential amino acids residues. What is specific, the sequence of amino acids in catalytic centre of all tyrosine phosphatases is conserved.
We add some information about bacterial tyrosine kinases in conclusion section.
Reviewer 3 Report
The review focuses on bacterial protein tyrosine phosphatases as promising targets for antimicrobial therapies. Despite of significant progress in identification of bacterial protein phosphatases responsible for bacterial virulence, development of inhibitors of such phosphatases is on early stage now. Thus, the review may encourage researchers to pay attention to this field. The review can be accepted after following minor issues be addressed.
1. Introduction. Were any other reviews in the field?
2. Table 1. Brief SAR discussion would be useful here.
3. Line 178. H in "Hydrogen peroxide" should be not capital.
4. 5.2. Inhibitors from natural sources. Structure of most important inhibitors mentioned here should be shown.
Author Response
Thank you so much for all the valuable comments and suggestions.
The other review in the field that we found are mentioned in a manuscript. It is reference 6,13,50 and 71,72. However, the most recent work is from year 2012. This is one of the main reason that we decided to prepare this paper, as there is not so many reviews on this topic.
We corrected the error in hydrogen peroxide.
We believe that SAR discussion of all presented inhibitors would be a great option for another review. Currently, we are afraid that we don’t have enough knowledge and materials to prepare it.
The structures of natural compounds are included in Table 1 as reference 54 and 58.
Round 2
Reviewer 1 Report
Dear Authors, I appreciated the efforts you made to improve the quality of the manuscript.
I had no further comments